



# Contribution of satellite sea surface salinity to the estimation of liquid freshwater content in the Beaufort Sea

Marta Umbert[1], Eva De Andrés[1,2], Maria Sánchez[1], Carolina Gabarró[1], Veronica González-Gambau[1], Aina García[1], Estrella Olmedo[1], Roshin P Raj[3], Jiping Xie[3], and Rafael Catany[4,5]

[1]Barcelona Expert Center on Remote Sensing, Institut de Ciències del Mar, CSIC, 08003 Barcelona, Spain.
[2]Department of Applied Mathematics, Universidad Politécnica de Madrid, Madrid, 28040, Spain.
[3]Nansen Environmental and Remote Sensing Center (NERSC) and Bjerknes Center for Climate Research, Bergen, 5007, Norway.
[4]ARGANS Ltd., Plymouth Science Park, 1 Davy Road, PLYMOUTH, PL6 8BX
[5]Albavalor, SL. Calle Catedratico Dr. D. Agustin Escardino Benlloch, 9, Parque Cienti. 46980, Paterna (Valencia). España

**Correspondence:** Marta Umbert (mumbert@icm.csic.es)

**Abstract.**
The hydrography of the Arctic Ocean has experienced profound changes over the last two decades. The sea-ice extent
has declined more than 10% per decade, and its liquid freshwater content has increased mainly due to glaciers and sea ice
melting. Further, new satellite retrievals of Sea Surface Salinity in the Arctic might contribute to better characterizing the
freshwater changes in cold regions. That is because ocean salinity and freshwater content are intimately related such that an
increase/decrease of one entails a decrease/increase of the other. In this work we evaluate the freshwater content in the Beaufort
Gyre, using surface salinity measurements from the satellite radiometric mission Soil Moisture and Ocean Salinity (SMOS)
and reanalysis salinity at depth. We estimate the freshwater content from 2011 to 2019 in the Beaufort Gyre and validate the
results with in-situ measurements. The results highlight the underestimation of the freshwater content using reanalysis data
in the Beaufort Sea and a clear improvement in the freshwater content estimation when adding satellite sea surface salinity
measurements above the mixed layer. The improvements are significant, especially in areas close to ice melting. Our research
demonstrates how remotely sensed salinity can assist us in better monitoring the changes in the Arctic freshwater content and
improving our understanding of a key process that is creating subtle density differences that have the potential to change the
global circulation system that regulates Earth's Climate.
**1   Introduction**
The Arctic has experienced rapid changes in recent years due to rising temperatures (Rantanen et al., 2022). Along with the
Arctic water cycle intensification, the sea ice cover is getting younger, thinner, and more mobile (Morison et al., 2012; Moore
et al., 2021). Retreating and decreasing sea ice cover, melting ice sheets and glaciers, and increasing Arctic river discharges



have led to a freshening of the upper Arctic Ocean (Haine et al., 2015; Solomon et al., 2021). The liquid freshwater content
(referred to as FWC) within the upper Arctic Ocean is maintained through the contributions of various significant factors.
These factors include river discharge, which accounts for approximately 40% of the FWC. The substantial inflow of relatively
fresh Pacific waters through the Bering Strait constitutes another vital component, contributing around 30% to the FWC.
Additionally, the balance between precipitation and evaporation plays a crucial role, with a net effect of approximately 25% on
the FWC. (Serreze et al., 2006; Timmermans and Marshall, 2020). These freshwater inflows play a vital role in maintaining the
halocline stratification of the Arctic Ocean, which serves as a protective barrier for the Arctic sea ice cover from the influence
of the warmer, deeper Atlantic waters.
At the heart of this Arctic climate system lies the Beaufort Gyre, a large swirling circulation cell north of the Beaufort Sea.
Since 1997, high atmospheric pressure has triggered strong anticyclonic winds over the Beaufort Gyre area (Lenton et al.,
2019). These winds drive the powerful clockwise circulation of the Beaufort Gyre. The gyre contains an enormous reservoir
of sea ice and freshwater from northern rivers (mainly Mackenzie and Yukon) and the Bering Strait (Proshutinsky et al., 2015;
Armitage et al., 2020). The FWC of the Beaufort Gyre has increased by 40% in the last two decades (McPhee et al., 2009;
Solomon et al., 2021). The variability of freshwater fluxes from the Arctic has the potential to affect global climate via the
global thermohaline circulation (Rahmstorf, 2000; Zhang et al., 2021; Årthun et al., 2023), as well as the ocean heat content
and biogeochemical cycles (Li et al., 2009). Between 2012 and 2016, the greatest and fastest change in salinity has been
reported (Sgubin et al., 2017), with the potential that subpolar North Atlantic convection collapse, resulting in rapid North
Atlantic cooling (Holliday et al., 2020).
Traditionally, the Arctic Ocean's FWC has been estimated using in situ hydrographic measurements. However, limited
spatiotemporal sampling and the coverage of in situ measurements pose a significant challenge to monitoring the FWC. In the
last decade, satellite data such as altimetry (e.g. Sea Surface Height from CryoSat-2) and gravimetry (e.g. bottom pressure from
GRACE), along with in situ observations and model reanalysis outputs, have been used to compute FWC estimations (Morison
et al., 2012; Armitage et al., 2016; Solomon et al., 2021). The difference between sea surface height anomalies derived from
altimetry measurements and ocean bottom pressure anomalies obtained from GRACE primarily represents the integrated steric
sea level variations across the water column. However, salinity is still considered a better indicator for estimating Arctic
freshwater (Fournier et al., 2019). In the Arctic Ocean with these cold ocean temperatures, the steric, or density, component
of sea level is primarily due to halosteric (salinity-induced) changes in the salinity of the upper ocean. Thereby, changes in
FWC are predominantly governed by alterations in salinity conditions, emphasizing the significant influence of salinity-related
changes on the sea level dynamics in the Arctic Ocean (Raj et al., 2020). This implies that salinity is the most natural variable
for investigating FWC as it directly describes the increases or decreases of freshwater in the ocean (Köhl and Serra, 2014; Tang
et al., 2018).
Since 2010, the retrieval of Arctic sea surface salinity (SSS) from microwave radiometric measurements obtained by satel-
lites such as Soil Moisture and Surface Salinity (SMOS; launched in 2009) (Reul et al., 2020), Aquarius (operational from 2011
to 2015) (Lagerloef, 2012), Soil Moisture Active Passive (SMAP; launched in 2019) (Tang et al., 2017), and future Copernicus
Imaging Microwave Radiometer (CIMR) satellite (Tang et al., 2017), has revolutionized the monitoring of the global water cy-





cle. The surface salinity observations allow us to improve the monitoring of the sea ice decline and river discharge impact and
analyze the water influx to the Arctic Ocean (Kilic et al., 2018). These satellites provide SSS estimates with a temporal repeat
cycle of approximately 1 day and an effective spatial resolution of 50 km in the seasonally ice-free areas of the Arctic Ocean
(Martínez et al., 2022). Due to low seawater temperatures of high latitudes, compared to lower latitudes, L-band brightness
temperatures in polar oceans exhibit lower sensitivity to changes in salinity. Consequently, inherent uncertainties are associated
with retrieving SSS in the Arctic from these satellite missions (Olmedo et al., 2018; Xie et al., 2019). However, significant ad-
vancements in retrieval algorithms have been made, leading to the development of specially tailored Arctic products (Martínez
et al., 2022) that have paved the way for integrating sea surface salinity data into studies focused on the Arctic FWC (Fournier
et al., 2019; Hall et al., 2021; Umbert et al., 2021; Hall et al., 2023).
In this work we evaluate the FWC in the Beaufort Gyre, using a satellite-derived Arctic SMOS SSS product with salinity
within the water column from TOPAZ4b reanalysis. By exploiting the capabilities of SMOS and merging its SSS observations
with salinity from reanalysis models, we aim to enhance our understanding of the distribution and dynamics of FWC in the
Beaufort Gyre region.

## 2 Data and Methods

### 2.1 Satellite data

The data utilized for conducting this analysis is the BEC SMOS Arctic Sea Surface Salinity product v3.1, described in
(Martínez et al., 2022). These salinity maps are generated on a daily basis, using a 9-day running mean, in an EASE 2.0
grid of 25 km. Data closer to 100 km to the coast lacks information as these pixels are expected to have low quality due to
land-sea contamination. The product is freely distributed from the Barcelona Expert Center website at http://bec.icm.csic.es/,
with the corresponding DOI number https://doi.org/10.20350/digitalCSIC/12620. Additionally, the data is also accessible on
the Digital CSIC server at https://digital.csic.es/handle/10261/219679.
The major advantage of this specially tailored product for the Arctic Ocean is the improvement of the effective spatial
resolution that permits better monitoring of the mesoscale structures larger than 50 km. This finer spatial resolution is one
of the main advantages of this product, as evidenced by the spatial-spectral analysis performed in (Martínez et al., 2022).
Therefore, this product is suitable for studying Arctic Ocean SSS processes and dynamics.
Daily sea ice concentration (SIC) estimates from the OSI-SAF Sea Ice Climate Change Initiative product OSI-430-b EU-
METSAT Ocean and Sea Ice Satellite Application Facility, Darmstadt, Germany (2019) were obtained from the Satellite
Application Facility on Ocean and Sea Ice (http://www.osi-saf.org/).

### 2.2 Reanalysis data

The TOPAZ system, developed at the Nansen Environmental and Remote Sensing Center (NERSC) and operated by the
Meteorological Institute of Norway, is an operational coupled ice-ocean data assimilation system specifically designed for the



Arctic Ocean. This system utilizes the HYCOM-CICE model with a resolution of 10 km across the entire Arctic region and
employs the Ensemble Kalman Filter (EnKF) technique with 100 dynamical members to assimilate all available ocean and sea
ice observations jointly (Xie et al., 2017).
We make use of the monthly outputs from the current version of TOPAZ system-TOPAZ4b reanalysis, spanning the years
2011-2019. Our focus is on the salinity variable, which is available at 40 vertical levels, ranging from surface to bottom.
The atmospheric forcing fields used in the TOPAZ4b are obtained from the ECMWF (European Centre for Medium-Range
Weather Forecasts). The HYCOM-CICE model is run on a daily basis, providing a 10-day forecast with an average of 10
ensemble members for the 3D physical ocean variables. Weekly data assimilation is performed to generate a 7-day analysis
using an ensemble average. It is important to note that this version TOPAZ4b incorporates the assimilation of the same SMOS
SSS product used in this study, as presented by (Xie et al., 2023), as well as other variables such as sea surface temperature,
sea ice concentration, salinity and temperature profiles, sea level anomaly, surface irradiance data, and sea ice thickness.
The output products of the TOPAZ4b are interpolated onto a grid with a resolution of 12.5 km at the North Pole, equivalent
to 1/8 degree in mid-latitudes. The interpolation is performed on a polar stereographic projection, and the hybrid vertical layers
are interpolated onto 40 fixed levels from the surface to 4000 m depth. These products serve as both near real-time forecast and
reanalysis products, contributing to the activities of the Copernicus Marine Services Arctic Monitoring and Forecasting Center
(Arctic MFC).
**2.3  *In-situ* data**
We utilize the FWC gridded data obtained from the Beaufort Gyre Exploration Project (Proshutinsky et al., 2009) to validate
the estimates that we present. They compute the FWC in the region, from 70°N to 80°N and 130°W to 170°W, where the water
depths exceed 300 meters. The data collected from CTD (conductivity-temperature-depth), XCTD (eXpendable Conductivity-
Temperature-Depth), and UCTD (Underway Conductivity-Temperature-Depth) profiles obtained between July and October
each year are used.
The in-situ FWC estimations are derived from salinity profiles and are optimally interpolated onto a 50-kilometer square grid,
providing insights into the FWC variability within the region. These maps cover the period from 2003 to 2020. Additionally,
uncertainties associated with each grid cell are determined using the optimal interpolation technique described in (Proshutinsky
et al., 2009).
**2.4  Freshwater content calculation**
We have computed the FWC combining SMOS SSS and in-depth ocean salinity from the TOPAZ4b reanalysis in the Beaufort
Sea during the 2011-2019 period. We have computed the FWC using the classical relation (Haine et al., 2015; Proshutinsky
et al., 2019):





$$FWC = \int\limits_{Z=0\,\mathrm{m}}^{Z(S_{\mathrm{ref}})} \frac{S_{\mathrm{ref}} - S(z)}{S_{\mathrm{ref}}} \, dz; \quad S_{\mathrm{ref}} = 34.8\,\mathrm{psu} \tag{1}$$
The FWC computation used SMOS SSS measurements in the pixels where the satellite has coverage, excluding ice-covered
ocean areas, from the ocean surface (the first TOPAZ4b layer) down to the mixed layer depth (MLD). In other cases, FWC
computation used TOPAZ4b salinity. Toole et al. (2010) showed that the MLD in that area is ∼22 meters, with a seasonal
variability of ∼8 meters based on the results from in-situ CTD and ice-tethered profilers. As TOPAZ4b has predefined layers,
we try three different TOPAZ4b layers as the depth of the mixed layer: 16, 25, and 29 meters, to assess the uncertainty
associated with using a constant value as the MLD through the year and the area. This generates an uncertainty that has an
impact on the FWC estimates because the MLD has a seasonal and inter-annual variability (Toole et al., 2010).
**3  Results and Discussion**
In our analysis, we exploited the data obtained from the SMOS microwave satellite. It is important to note that the coverage of
SSS data from microwave satellites is limited in the presence of sea ice (Figure 1). During periods of sea ice melting, a larger
area of the ice-free ocean becomes observable, enabling SMOS to detect SSS. These measurements provide valuable insights
into the variability of the FWC of the region resulting from recent ice melting. Other processes associated with surface salinity
in the Arctic region that SMOS potentially can detect are precipitation, river runoff, and circulation patterns such as currents,
and eddies that transport water masses with different salinity characteristics. Furthermore, SMOS satellite can also potentially
detect high saline waters surfaced due to vertical mixing processes.



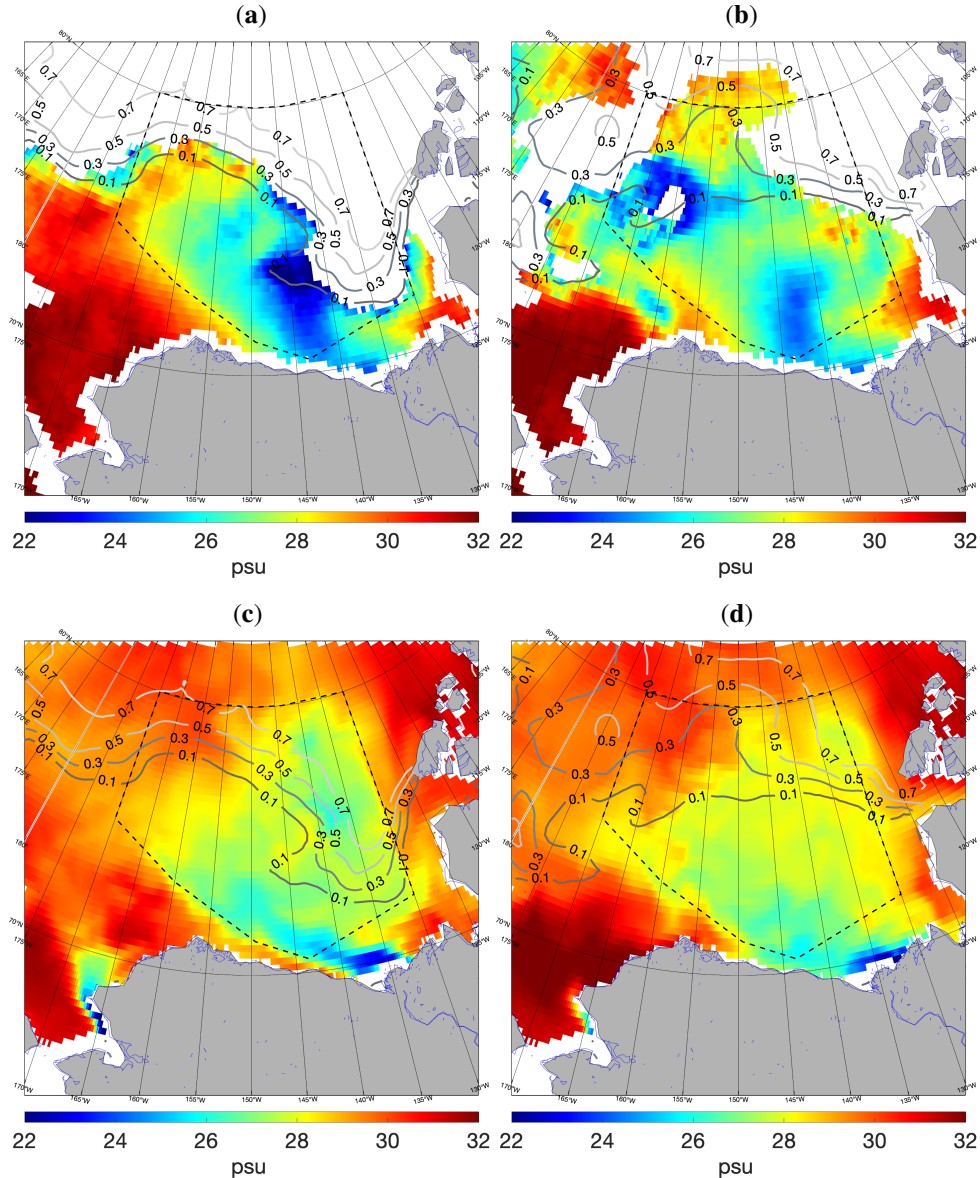

**Figure 1.** Mean SMOS SSS for September 2011 (a) and September 2016 (b). Mean uppermost salinity level of TOPAZ4b for September 2011 (c) and September 2016 (d). The average sea ice concentration contours for September 2011 and 2016 provided by OSISAF are overlayed. The study area of the Beaufort Gyre is in black dashed lines.

Figure 1 displays the monthly averaged surface salinity observed by SMOS during September 2011 and September 2016
(panels a and b, respectively). The surface salinity (first layer) from the TOPAZ4b reanalysis for the same period is shown in
panels c and d. The satellite data exhibits lower salinity values than those resolved by the reanalysis. The reanalysis captures low
salinities in the Mackenzie River plume, however, miss the low salinities in the center of the Beaufort Gyre, which may have its





origin from the melting of sea ice, and/or may be associated with fresh waters from rivers such as the Ob Lena and the Yenisei
in the Eurasian Basin, transported into this region Proshutinsky et al. (2009); Hall et al. (2023). Note that even if TOPAZ4b
reanalysis assimilates SMOS SSS, the resulting surface salinity does not seem to reproduce the same SSS dynamics as seen
by SMOS. As indicated by the contours of sea ice concentration overlayed in the figure, there are areas with SMOS salinity
data but not free of ice coverage. This is because the SMOS SSS data is a monthly average of daily products generated using a
9-day running mean. Therefore, these areas represent regions where ice has recently retreated, leaving behind melt waters. The
satellite data appears to capture the freshwater input resulting from ice retreat (Eva De-Andrés and Gabarró, 2023).

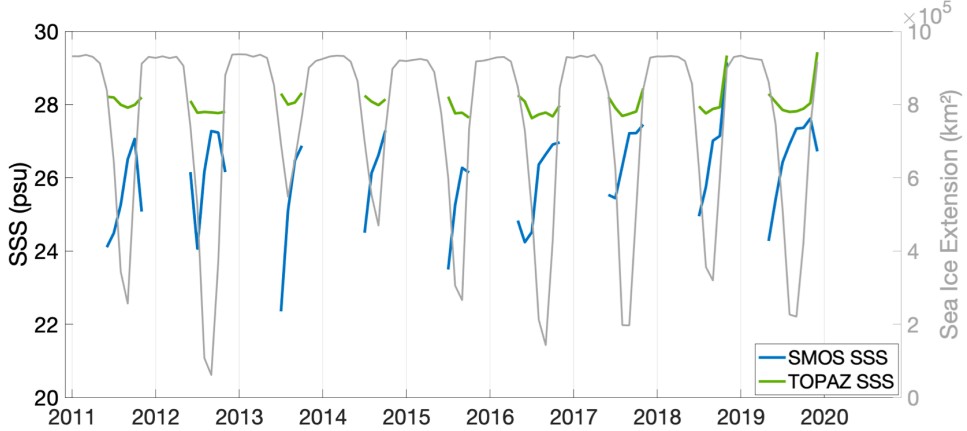

**Figure 2.** Temporal evolution of mean SMOS SSS, TOPAZ4b SSS (in the same pixels as SMOS), and OSISAF sea ice extension during
2011-2019 in the Beaufort Gyre.

The temporal evolution of the satellite and reanalysis surface salinity (Figure 2), further highlights high reanalysis salinities
in the region. The seasonal variability in the reanalysis salinities (green line) is very low, while SMOS SSS (blue line), captures
both fresh waters from the ice melting during early summer, and high salinities during the ice formation in fall. When the ice
coverage decreases during the spring and summer months, satellite salinity reveals a noticeably lower salinity than TOPAZ4b
(salinity values ranging from 1 to 4 less on average, depending on the period). Even if TOPAZ4b assimilates SMOS SSS
information, the surface salinity in the reanalysis is still far from the satellite observations, mainly due to the excessively low
weight assigned to SMOS measurements, and an excessive SSS relaxation process to the World Ocean Atlas (WOA18) SSS in
the assimilation scheme.

### 3.1  Freshwater content using salinity

In the Beaufort Sea region, we observed that the SSS obtained from SMOS data tends to be fresher compared to the sur-
face salinity provided by the TOPAZ4b reanalysis model (Figure 2). This discrepancy in salinity motivates the necessity of
incorporating SMOS SSS up to the MLD to estimate FWC in this key region of the Arctic Ocean.




We determine the FWC (Section 2.4), within the Beaufort Gyre region, defined from 70°N to 80°N and 130°W to 170°W,
in areas where water depths exceed 300 m, to emulate the area of the in-situ measurements (Section 2.3). To calculate the
FWC by merging SMOS SSS and TOPAZ4b salinity, we combine the salinity data from the TOPAZ4b reanalysis at various
depths with the SMOS SSS values for the layers above the MLD. This methodology is detailed in Section 2. By integrating the
remotely sensed salinity, we aim to obtain a more accurate estimation of the FWC within the Arctic Ocean.

**Figure 3.** Mean freshwater content using only TOPAZ4b (a,d), TOPAZ, and SMOS SSS on the first 16 meters (b,e) and freshwater content difference (c,f) for September 2011 (top row) and September 2016 (bottom row). The freshwater content difference is computed as the freshwater content from TOPAZ4b salinity minus the freshwater content from TOPAZ4b adding SMOS up to 16 meters.

Figure 3 presents the FWC estimates in September 2011 and 2016, using only reanalysis salinity (a and d), and those by
introducing SMOS SSS up to the layer of 16 meters in TOPAZ4b (b and e). Similar results but with higher FWC are found when
SMOS SSS is added up to 25 or 29 meters (spatial map not shown, but results are found in Table 1 and Figure 4). Compared to
the reanalysis-only data, the FWC values are higher when SMOS information is integrated into the TOPAZ4b data. Figure 3



c and f presents the difference in FWC between the TOPAZ4b-only estimates and the one which incorporates the SMOS SSS
information up to the upper 16 m (similar patterns with higher differences are found for 25 and 29 m, not shown). The impact
of including SMOS SSS data in FWC computation is particularly pronounced in regions affected by sea ice melting (Figure 3
c and f). These regions are characterized by dynamic changes in salinity due to the mixing of ice melt-induced freshwater with
the underlying seawater. By incorporating SMOS SSS information in these areas, we expect higher values of FWC estimates,
as SMOS observations reflect fresher surface waters (Figures 1 and 2).

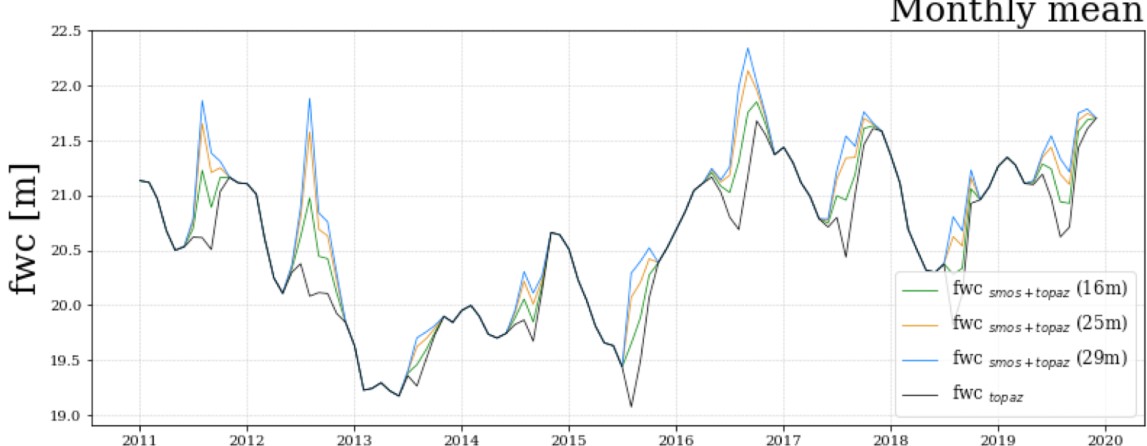

**Figure 4.** Temporal evolution of freshwater content in the Beaufort Gyre using TOPAZ4b salinity (black line), and adding SMOS SSS up to
16 m (green line), 25 m (orange line), and up to 29 m (blue line).

The mixed layer depth of the region is in the range of 20 m (Toole et al., 2010), and when introducing SMOS SSS information
within the mixed layer (up to different TOPAZ4b layers 16, 25, 29 m, see Section 2.4), higher FWC values are obtained (Figure
4 and Table 1). This indicates that incorporating SMOS SSS data produces an increase in the estimation of FWC, a mean
increment on average of approximately 3-6% in FWC values in the Beaufort Gyre. However, if we consider only the ice-free
region (area seen by SMOS), the increase in FWC can reach up to 6-10% (Table 1). Table 1 provides evidence that during
summer-autumn months (July, August, September, and October), the estimated FWC in the Beaufort Gyre and the ice-free
area is very similar.
In the climate model used in Rosenblum et al. (2021), the bias in surface salinity was found to be mainly attributed to
unrealistically deep vertical mixing in the model, creating a surface layer that is saltier than observed. This bias can affect
the accuracy of FWC estimates, leading to an underestimation compared to in-situ measurements. Other reasons that can
also explain why reanalysis models may underestimate FWC estimates as compared to estimates from in-situ measurements.
There are model biases and limitations inherent in the reanalysis due to simplifications and approximations in their numerical
representations of complex Arctic Ocean processes Heuzé et al. (2023). Reanalysis models may not fully capture or accurately
parameterize all the relevant physical processes as the ones related to freshwater inputs, such as precipitation, runoff, or ice
melt, which may not be adequately represented, resulting in underestimated FWC estimates. Our results suggest that there is



**Table 1.** Yearly freshwater content mean for months of July, August, September, and October, and freshwater content in the ice-free region using only TOPAZ4b salinity, and adding SMOS SSS up to 16, 25, and 29 meters depth for each of the years from 2011 to 2019. Units are meters.

| fwc / fwc$_{ice\text{-}free}$ | TOPAZ4b Only | SMOS 16 m. | SMOS 25 m. | SMOS 29 m. |
|---|---|---|---|---|
| 2011 | 20.44 / 20.71 | 20.81 / 21.71 | 21.11 / 22.44 | 21.27 / 22.82 |
| 2012 | 20.07 / 19.81 | 20.64 / 20.67 | 21.05 / 21.27 | 21.27 / 21.58 |
| 2013 | 19.18 / 18.47 | 19.37 / 19.27 | 19.55 / 20.06 | 19.64 / 20.50 |
| 2014 | 19.59 / 19.89 | 19.79 / 20.63 | 19.98 / 21.27 | 20.09 / 21.63 |
| 2015 | 19.22 / 19.90 | 19.60 / 20.79 | 19.89 / 21.49 | 20.07 / 21.88 |
| 2016 | 20.98 / 20.85 | 21.43 / 21.71 | 21.76 / 22.30 | 21.94 / 22.61 |
| 2017 | 20.83 / 21.34 | 21.16 / 21.93 | 21.43 / 22.40 | 21.59 / 22.67 |
| 2018 | 20.23 / 20.09 | 20.51 / 20.70 | 20.52 / 21.18 | 20.85 / 21.47 |
| 2019 | 21.01 / 21.09 | 21.34 / 21.62 | 21.59 / 22.03 | 21.73 / 22.27 |

room for further improving the freshwater influx from sea ice in the TOPAZ4b reanalysis system and is expected to be corrected
in the next release.
**3.2 Validation using in-situ FWC estimates**
In this section, we use the in-situ dataset from the Beaufort Gyre Experiment Project (Section 2.3) to validate the FWC
estimations using salinity from satellite and reanalysis. To compare with these estimations, we linearly interpolate the FWC
estimates using SMOS surface salinity data and column water salinity information from the TOPAZ4b reanalysis onto the
same 50 km grid and time period. Figure 5 depicts the in-situ FWC measurement for the year 2011 (Figure 5a), as well as the
estimation solely based on TOPAZ4b (Figure 5b), and SMOS up to 25 meters (Figure 5c). It is evident from the figures that the
FWC only with TOPAZ4b significantly underestimates the amount of FWC with respect to the in-situ data. Introducing SMOS
information brings the FWC estimation closer to the in-situ estimates (Figure 5d and e), decreasing the negative bias in the
pixels where SMOS information was available (Figure 5f). It is worth noting that the estimates were better where the SMOS
observations were used.





**Figure 5.** Yearly mean for 2011 of freshwater content [meters] from (a) in-situ measurements interpolated into a 50 km grid by the Beaufort Gyre Experiment Project (Proshutinsky et al., 2009), (c) only TOPAZ4b salinity, and (e) SMOS up to 25 meters and TOPAZ4b salinity. (b) The error associated with the in-situ FWC estimation related to the optimal interpolation scheme (Proshutinsky et al., 2009). Difference between FWC estimations using (d) TOPAZ4b salinity, and (f) SMOS up to 25 meters and TOPAZ4b salinity against in-situ.



The FWC obtained using only reanalysis salinity data underestimates FWC from in-situ measurements. This fact is already
pointed out by several studies using different ocean models (Hall et al., 2022). The inclusion of SMOS SSS data within the
MLD enhances the estimation of FWC, leading to higher values, especially in regions affected by sea ice melting. Our findings
emphasize the valuable contribution of SMOS SSS data in enhancing our comprehension of freshwater dynamics in the studied
area, as well as the valuable information that satellite salinity measurements can provide in monitoring the surface freshwater
flux in the region during these months.

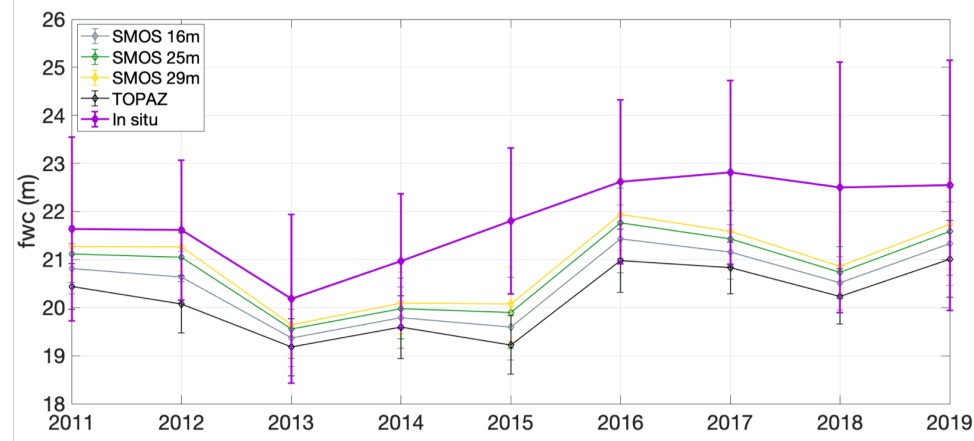

**Figure 6.** Temporal evolution of mean freshwater content (between July and October) in the Beaufort Gyre computed using only TOPAZ4b
(black), and TOPAZ4b with SMOS SSS until 16 (grey), 25 (green), and 29 (yellow) m depth, and from in-situ data (purple).

When introducing SMOS SSS data, the mean annual FWC estimates (between July and October) in the Beaufort Gyre region
exhibit a significant improvement compared to in-situ estimates (Figure 6). For example, the incorporation of SMOS SSS data
within the upper 25 m depth leads to a noteworthy 34.8% decrease in bias (Figure 7). Additionally, there is a notable 14.55%
increase in slope, indicating a better alignment between the FWC from SMOS estimates and the observed values from in-situ
measurements. Moreover, there is a non-negligible 4.08% increase in the coefficient of determination ($R^2$) (Figure 7). We
computed the percentage of increase/decrease as ((new value − initial value)/ initial value) x 100). This indicates an enhanced
level of agreement when computing the FWC values combining SMOS SSS and TOPAZ4b and those obtained from in-situ
measurements.





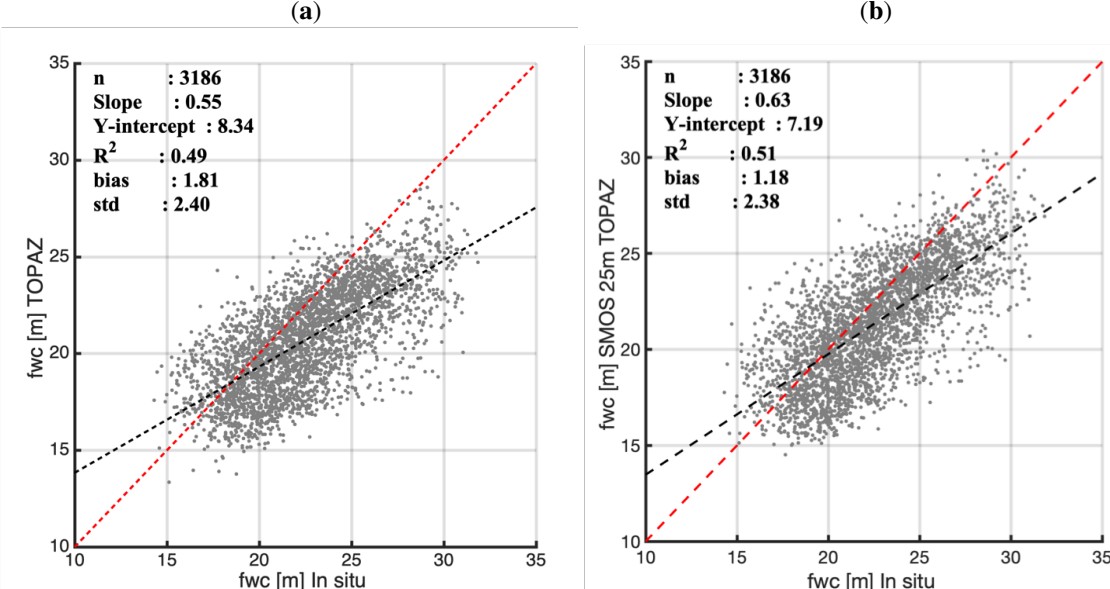

**Figure 7.** Scatterplot of mean yearly freshwater content at each point of the Beaufort Gyre since 2011-2019 from in-situ estimates against the freshwater content from (a) TOPAZ4b and from (b) TOPAZ4b and SMOS data in the first 25 m depth for the same period and resolution.

Table 2 presents the validation results of FWC estimates based on the salinity from the TOPAZ4b reanalysis, either alone or
by adding the surface salinity from SMOS down to the mixed layer depth at three different values of MLD using the FWC from
in-situ data. It is observed that the bias decreases when SMOS data is added in the upper layers. Typically, the bias decreases
by 30% when SMOS data is added within the first 16 m depth, and between 50 and 70% when information is added up to
25 and 29 m depth, respectively. Although the results show a significant improvement in terms of bias, the standard deviation
does not significantly change (+ or - 10%) when SMOS data is added (Figure 7 and Table 2). The standard deviation between
model-based and in-situ-based estimates have the same order of magnitude (1-3 meters) that the error of in-situ estimates due
to the optimal interpolation scheme applied (Proshutinsky et al., 2019).
Probably the dispersion remains stable since it is determined by the difference in structures that can be resolved between
interpolated in-situ measurements on one hand and a reanalysis that incorporates satellite data on the other. By adding SMOS
data, it could even lead to increased dispersions since SMOS salinity measurements have a finer spatial resolution, allowing
for the detection of in-situ unrevealed structures. Additionally, SMOS provides daily and integrated temporal resolution during
ice-free months, which contrasts with in-situ measurements which are point measurements conducted on ice-tethered drifts or
on sea ice masses that SMOS cannot measure. Overall, these findings demonstrate that incorporating SMOS SSS data within
the mixed layer depth significantly improves the accuracy of FWC estimates. The reduced bias, increased slope, and improved
coefficient of determination suggest a better representation of FWC when compared to in-situ estimates.





**Table 2.** Bias and standard deviation of yearly mean FWC using only TOPAZ4b salinity, and adding SMOS SSS up to 16, 25, and 29 m depth against in-situ FWC estimates for years from 2011 to 2019.

| BIAS / STD | TOPAZ4b Only | SMOS 16 m. | SMOS 25 m. | SMOS 29 m. |
|:---:|:---:|:---:|:---:|:---:|
| 2011 | 1.28 / 1.64 | 0.86 / 1.63 | 0.55 / 1.70 | 0.38 / 1.76 |
| 2012 | 1.82 / 2.16 | 1.25 / 2.28 | 0.86 / 2.44 | 0.64 / 2.54 |
| 2013 | 0.99 / 1.63 | 0.87 / 1.72 | 0.75 / 1.85 | 0.68 / 1.93 |
| 2014 | 1.42 / 1.99 | 1.27 / 2.10 | 1.12 / 2.23 | 1.04 / 2.33 |
| 2015 | 2.63 / 1.96 | 2.17 / 1.91 | 1.82 / 1.97 | 1.62 / 2.04 |
| 2016 | 1.68 / 2.40 | 1.21 / 2.21 | 0.88 / 2.14 | 0.70 / 2.12 |
| 2017 | 2.02 / 2.39 | 1.70 / 2.30 | 1.46 / 2.29 | 1.32 / 2.29 |
| 2018 | 2.52 / 3.33 | 2.20 / 3.21 | 1.95 / 3.15 | 1.81 / 3.12 |
| 2019 | 1.66 / 2.96 | 1.39 / 2.92 | 1.18 / 2.92 | 1.06 / 2.93 |

## 4 Conclusions

Ongoing improvements in sea surface salinity (SSS) retrievals have the potential to significantly advance our understanding of freshwater changes in the Arctic. The Arctic freshwater system is complex and understanding its dynamics is crucial for studying the impacts of climate change in the region. This work computed the freshwater content by combining SMOS sea surface salinity data and ocean salinity in depth from the TOPAZ4b reanalysis for the period of 2011-2019. To validate our results, we compared them to FWC estimates derived from in-situ conductivity-temperature-depth measurements in the Beaufort Sea region generated by the Beaufort Gyre Experiment Project (Proshutinsky et al., 2009).

The accuracy of FWC estimates from reanalysis models is an ongoing research topic, and efforts are continuously made to improve the models and their representations of FWC. Despite this, when using only TOPAZ4b salinity data, the computed FWC underestimates the values obtained from in-situ measurements. However, incorporating SMOS SSS data from the surface down to the mixed layer depth of 29 m results in an average increase of up to 10% in the FWC values. This demonstrates the capability of SMOS SSS data for capturing the spatial and temporal variations in FWC, especially in regions where sea ice melting plays a significant role in the overall freshwater balance and the importance of assimilating SSS on models.

It is important to note that the choice of the surface layer thickness, where we introduce SMOS SSS data, affects the results. We found that introducing the SMOS SSS data in the mixed layer depth of 25-29 m provides the best agreement with in-situ measurements. We need better monitoring of the depth of the mixing layer in order to more accurately estimate the true impact of assimilating SMOS data in this type of analysis. Our results suggest that more weight should be given to the SMOS SSS measurements in the assimilation into the TOPAZ4b model and routinely integrated into Arctic oceanographic models. Overall, by combining SMOS SSS and TOPAZ4b data, along with careful consideration of the surface layer thickness, we have improved the accuracy of FWC estimates compared to using reanalysis data alone.



Finally, in agreement with previous authors (e.g. Tang et al. (2018); Fournier et al. (2020); Hall et al. (2023)), this work
highlights the value of SSS for studying freshwater variability in the Beaufort Sea. Ongoing improvements in SSS retrievals
can significantly advance our understanding of Arctic freshwater distribution. Integrating and analyzing SSS data from various
sources, including satellite remote sensing, in-situ measurements, and numerical models, enables a comprehensive under-
standing of the Arctic freshwater system. This integrated approach could allow for the identification of patterns, trends, and
anomalies in SSS, which can provide valuable insights into the drivers and impacts of freshwater changes in the Arctic in the
broader context of climate change and global ocean dynamics.
*Author contributions.*
MU: Conceptualization, investigation, methodology, formal analysis, validation, writing - original draft. EDA: Investigation,
methodology, formal analysis, review, and editing. MS: Investigation, methodology, review, and editing. CG: Funding acqui-
sition, investigation, review, and editing. VGG: Review, editing. AG: Data curation. EO: Review and editing. JX: Review and
editing. RC: Project management, review, and editing.
*Competing interests.*  No competing interests are present.
*Acknowledgements.*  This project was founded by Marie Skłodowska-Curie Grant Agreement No. 840374. E. De Andrés is funded by Mar-
garita Salas Grant No. UP2021-035 under the Next Generation EU program and supported by the MCIN/AEI project PID2020-113051RB-
C31. We also received funding from the AEI with the ARCTIC-MON project (PID2021-125324OB-I00) and from the ESA Arctic+ Salinity
project (AO/1-9158/18/I-BG) and Arctic+ SSS CCN (4000125590/18/I-BG). This work represents a contribution to the CSIC Thematic In-
terdisciplinary Platform PTI-POLARCSIC and PTI-TELEDETECT and is supported by the Spanish government through the "Severo Ochoa
Centre of Excellence" accreditation (CEX2019-000928-S).



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
