# Peer review of "Contribution of satellite sea surface salinity to the estimation of liquid freshwater content in the Beaufort Sea"

_EGUsphere, 2023_

## Referee Comment (RC1)

**Review**

I conducted a thorough review of the manuscript titled "Contribution of satellite sea surface salinity to the estimation of liquid freshwater content in the Beaufort Sea" by Marta Umbert et al. In summary, these findings highlight that the incorporation of SMOS SSS data while employing TOPAS4b reanalysis data into the calculation of freshwater content in the Beaufort Gyre region during 2011-2019 summer months. Three different depths were used to combine SMOS SSS in freshwater content (FWC) depending on a constant mixed layer depth which significantly enhances the accuracy of FWC estimates. This enhancement is manifested through the reduction of bias, an increased slope, and improved coefficient of determination when compared to in-situ estimates.

General Recommendations:
- Maintain consistency in the use of "in-situ" terminology, either hyphenated without italics or not hyphenated with italics, as long as it is applied uniformly throughout.
- Strive for conciseness and directness in many areas of the text. Specific values can replace vague sentences to enhance precision.
- Clarify a major concern regarding the measurement of freshwater content. It should be noted that mixed layer depth can be calculated, and the study is concerned with the amount of freshwater within this depth while testing different constant MLD values. The justification for this approach, represented by adding SMOS SSS data up to 16 m (green line), 25 m (orange line), and up to 29 m (blue line), should be provided. Additionally, address the potential for overestimation of FWC due to SMOS measurements being confined to the melting season, which can create a freshwater film at the surface.
- I have reservations to these results, one of which is related to alias biasing of in-situ data based on their horizontal and vertical resolution. The FWC product is integrated which can introduce more bias and thus may not represent 'true FWC' without further explanation of the product or related works. Emphasize that the estimation of FWC remains subject to limitations when insufficient data is available, as comparisons to direct in-situ measurements allows for validation of salinity but not the integration of freshwater.
- Acknowledge concerns about the region tested, particularly the influence of downwelling on freshwater accumulation, which could be the reason for the improvement using SMOS SSS to 29 m depth. Concluding that combining SMOS SSS and reanalysis data could enhance the entire Arctic's FWC should be cautioned as this has not been tested in your research while it may be a useful avenue for future exploration.

**Abstract**

Authors specify their use of satellite data to better assess SSS in the Arctic. Such retrievals are known to have substantial limitations and large errors near the sea ice edge and in cold waters due to the L-band retrievals becoming less sensitive to salinity at cold sea surface temperatures. Can the authors justify this in text?

Line 4: "Sea Surface Salinity" does not need capitalization

Line 5-6 "That is because… increase/decrease of one entails a decrease/increase of the other." The authors should specify this point for clarity. As in "an increase (decrease) of salinity entails a decrease (increase) of freshwater content". "That is because" is not needed.

Line 8: Specify type of reanalysis product you are using as your study does not analyze more than one reanalysis product.

Line 11: "The improvements are significant, especially in areas close to ice melting."
Too vague, give a more quantitative value from results to back this.

Line 13-14: "Our research demonstrates….that regulates Earth's Climate."
This sentence should be rewritten for clarity. Some areas are vague ("a key process", which one?) and the research does not address the implications towards the global circulation system that regulates the Earth's Climate. Add further details in the introduction to justify this connection.

**Introduction**
- Address the significance of atmospheric conditions and climate patterns in relation to the retention of FWC in the Beaufort Gyre, which currently lacks explanation in the introduction.
- It might be best to describe what liquid freshwater content is since its not just a layer of freshwater on top of salt water, but a ratio of salt content that is lower than a certain standard. The salinity reference is also not agreed upon between scientists and may need to be defended on your part for why you chose that salinity reference (simply stating that you match the salinity reference with Proshutinsky et al. (2009) as you compare your results to their gridded FWC. It would also be beneficial to clarify the concept of FWC within the context of the Arctic. Emphasize that "freshwater" does not necessarily mean complete absence of salinity; there exists a salinity threshold that varies in Arctic research.

Line 17: Be more specific, the Arctic has experienced rapid changes more than just recent years, it has arguably been occurring over decades or at least since 2007. This statement could be improved by adding specific rate or timeframe that would enhance the importance of Arctic changes.

Line 21: "(referred to as FWC)"
Change to "(FWC)"

Line 23: Please provide a references for these contributions of FWC as it may differ between time periods or methodologies.

Line 28: This sentence is a bit misleading, the Beaufort Gyre itself isn't necessarily at the 'heart' of the Arctic's climate, its highly influenced by climatic systems and atmospheric processes. It is also located in the Beaufort Sea and may extend northward but this statement makes it sound like it is not in the Beaufort Sea.

Line 29-30: This is a personal opinion but will clarify for the reader: "…strong anticyclonic (counterclockwise) winds…powerful cyclonic (clockwise) circulation".

Line 32-33: This transition between sentences seems random, you started introducing the Beaufort Gyre then talk about the Arctic's freshwater flux influence on a global scale without stating how. You could emphasize that the Beaufort Gyre consists of a large portion of the entire Arctic Ocean's FWC and where the connection is between the Arctic and the thermohaline circulation comes in.

Line 35: I'm assuming "change" could be clarified as "increase"? This sentence also seems random and would benefit with explanation of the importance of the time between 2012 and 2016.

Line 40: Only within the last decade (2013-2023)?. Also, is sea surface height capitalized for a reason?

Line 47-49: It would be much simpler to state that FWC is the integral of salinity differences between measurement and a constant. I understand the way you are introducing different ways of measuring FWC but you don't describe what FWC really is upfront. This paragraph can be conveyed more directly and concise.

Line 56-57: One-day repeat cycle does not seem accurate. Data can be generated at daily intervals, but that is not the same as a satellite's repeat cycle. Typically, its 3-8 days unless you can clarify what you mean. I also believe Aquarius's spatial resolution is much greater.

Line 62: Change "sea surface salinity" to "SSS". Also in lines 70, 228.

Line 86: Change "resolution" to "spatial resolution"

Line 90: "ranging from surface to bottom". Can you be more specific on what 'surface' is in meters?

**Data**
Line 70: State the level of the satellite product (level 3 product?)

Line 71: Specify month and year range of SMOS data and which timeframe you took the data from here.

Line 80-81: You use the OSI-SAF acronym then the name but should it come after "Ocean and Sea Ice Satellite Application Facility" in parenthesis?

Line 96: Change "sea ice concentration" to SIC. Also in line 139.

Line 99: "..surface to 4000 m depth.", is this near surface or skin-surface? Might be best to note top layer in meters.

Line 105: Do the in-situ data exceed 300 meters as well or is this referring to the region's depths exceed 300 m defined by Proshutinsky. If the latter, then clearly define the depths that the in-situ measurements cover.

Line 109: Specify the time intervals that these data are provided or that you used (hourly, daily, monthly?)

Eq.1: Define equation symbols through text explanation.

Line 121: Justify the use of 16, 25, and 29 meters for the MLD. Can you use profile data to estimate the MLD? You mention using a constant value of MLD throughout the year but are only assessing melt season, which is confusing.

**Results**

Line 131: As a predominantly anticyclonic gyre, isn't it more characteristic for the Beaufort Gyre to have downwelling occur instead of upwelling as this line suggests (i.e. "surfaced")? Or are you discussing waters pulled from the surrounding water or characteristics in general?

Figure 1: Why these two years (2011 & 2016)? You mention the greatest and quickest change in salinity between 2012-2016, but do you choose 2011 since is the beginning of the period you are examining? Stating the uppermost level of TOPAS4b would be helpful in analyzing the difference between the skin-salinity that the satellite measures and the depth of what TOPAZ4b measures.

Line 137: Should the references be in parentheses?

Line 155-156: Repeat of the description of the Beaufort Gyre region as in lines 104-105, could further describe why you use the same region as Proshutinsky et al.

Figure 3. Note the acronym for freshwater content and its unit in the caption (i.e. FWC [m]). The colorbar for (c,f) seems to emphasize regions over or under a ~2psu difference, is there a reason for this or can the colorbar be changed to detail the region and make different values clearer?

Line 182: Parenthesis around reference.

Line 182-184: How does TOPAZ4b assimilate variables associated with freshwater inputs? Are there estimations on its certainty to capture these signals accurately?

**Section 3.2**
Figure 5. Label [b] colorbar as "FWC error [m]". Last sentence of caption is not complete.

Line 198: You mention "several studies" but only cite one, can you provide more studies to back this claim? Otherwise reword.

Figure 6.: Define more reasons why there are large discrepancies in FWC between in-situ data and other products. Could in-situ be overestimating due to lack of spatiotemporal coverage? The in-situ based FWC product is integrated so this could pose it's own errors or overestimation/underestimation.

Line 228: Change "sea surface salinity (SSS)" to "SSS"

Line 230: Change "…freshwater content.." to "FWC"

Lines 251-252: "This integrated approach could allow for the identification of patterns, trends, and anomalies in SSS.."

---

## Author Comment (AC1)

**Reply on RC1**

*Thank you for your careful and insightful review, all your suggestions have been addressed.*

**Review**
I conducted a thorough review of the manuscript titled "Contribution of satellite sea surface salinity to the estimation of liquid freshwater content in the Beaufort Sea" by Marta Umbert et al.
In summary, these findings highlight that the incorporation of SMOS SSS data while employing TOPAS4b reanalysis data into the calculation of freshwater content in the Beaufort Gyre region during 2011-2019 summer months. Three different depths were used to combine SMOS SSS in freshwater content (FWC) depending on a constant mixed layer depth which significantly enhances the accuracy of FWC estimates. This enhancement is manifested through the reduction of bias, an increased slope, and improved coefficient of determination when compared to in-situ estimates.

**General Recommendations:**
- Maintain consistency in the use of "in-situ" terminology, either hyphenated without italics or not hyphenated with italics, as long as it is applied uniformly throughout.
*Done*

- Strive for conciseness and directness in many areas of the text. Specific values can replace vague sentences to enhance precision.
*Done*

- Clarify a major concern regarding the measurement of freshwater content. It should be noted that mixed layer depth can be calculated, and the study is concerned with the amount of freshwater within this depth while testing different constant MLD values. The justification for this approach, represented by adding SMOS SSS data up to 16 m (green line), 25 m (orange line), and up to 29 m (blue line), should be provided. Additionally, address the potential for overestimation of FWC due to SMOS measurements being confined to the melting season, which can create a freshwater film at the surface.
*Certainly, as the reviewer points out, the calculation of Mixed Layer Depth (MLD) can be performed using in-situ profiles. However, for the calculation of FWC, we employ TOPAZ4 system that uses fixed depth levels. We use bibliographic estimates of typical MLD depths for the time of year and study area and conduct three experiments using the nearest fixed levels to these bibliographic MLDs. Through these three experiments, we can assess the uncertainty associated with the spatiotemporal variability of MLD. This is outlined in subsection 2.4.*

*We believe that we are not overestimating FWC. Instead, we think that the SMOS data introduce FWC originating from melting that is not accounted in TOPAZ4 system. When we provide an annual estimate of FWC (Figure 6), we specify that it only represents the mean for the months from July to October.*

- I have reservations to these results, one of which is related to alias biasing of in-situ data based on their horizontal and vertical resolution. The FWC product is integrated which can introduce more bias and thus may not represent 'true FWC' without further explanation of the product or related works. Emphasize that the estimation of FWC remains subject to limitations when insufficient data is available, as comparisons to direct in-situ

measurements allows for validation of salinity but not the integration of freshwater.- Acknowledge concerns about the region tested, particularly the influence of downwelling on freshwater accumulation, which could be the reason for the improvement using SMOS SSS to 29 m depth. Concluding that combining SMOS SSS and reanalysis data could enhance the entire Arctic's FWC should be cautioned as this has not been tested in your research while it may be a useful avenue for future exploration.

*Thank you for your input, we have introduced your points in the manuscript:*

*Section 3.2. 'It is worth considering that FWC estimates based on in-situ data also come with inherent biases, influenced by their horizontal and vertical resolution (Proshutinsky et al., 2009). The estimation of FWC remains an ongoing research topic due to the limitations posed by the scarcity of in-situ data available for producing these estimates.'*

*Section 3.2. 'A potential explanation for the higher improvement observed when using SMOS SSS data down to the 29-meter level, as opposed to the other experiments, could be associated with the impact of downwelling on freshwater accumulation in the Beaufort Gyre.'*

*Conclusions. 'This integrated approach could allow for the identification of patterns, trends, and anomalies in SSS, which can provide valuable insights into the drivers and impacts of freshwater changes in the Beaufort region, and hold promise for future exploration in the broader Arctic within the context of climate change and global ocean dynamics.'*

Abstract
Authors specify their use of satellite data to better assess SSS in the Arctic. Such retrievals are known to have substantial limitations and large errors near the sea ice edge and in cold waters due to the L-band retrievals becoming less sensitive to salinity at cold sea surface temperatures.
*In the abstract the authors state that: 'new satellite retrievals of Sea Surface Salinity MIGHT contribute to better characterizing the freshwater changes in cold regions'*

Can the authors justify this in text?
*The limitations of SSS in the Arctic are stated and referenced (Olmedo et al 2018 and Xie et al 2019) in the introduction.*

Line 4: "Sea Surface Salinity" does not need capitalization
*Done*
Line 5-6 "That is because… increase/decrease of one entails a decrease/increase of the other."
The authors should specify this point for clarity. As in "an increase (decrease) of salinity entails a
decrease (increase) of freshwater content". "That is because" is not needed.
*Done*
Line 8: Specify type of reanalysis product you are using as your study does not analyze more than one reanalysis product.
*Done*
Line 11: "The improvements are significant, especially in areas close to ice melting."
Too vague, give a more quantitative value from results to back this.
*Thank you, we have added 'The improvements are significant, with up to a 70% reduction in bias in areas near the ice melting'.*
Line 13-14: "Our research demonstrates….that regulates Earth's Climate."
This sentence should be rewritten for clarity. Some areas are vague ("a key process", which one?) and the research does not address the implications towards the global circulation system

that regulates the Earth's Climate. Add further details in the introduction to justify this connection.

*The key process that we mean is the changing freshwater content in the Arctic system due to glacier melting, increase river discharges and accumulation in the Beaufort Gyre.*

*The phrase now reads as: 'Our research demonstrates how remotely sensed salinity can assist us in better monitoring the changes in the Arctic freshwater content and improving our understanding this key process that is creating subtle density differences that have the potential to change ocean global circulation system that regulates Earth's Climate.'*

*We have added the following to the introduction: 'Changes in the Arctic hydrography directly affect conditions on the rest of the planet through feedback mechanisms and interactions with the northern hemispheric atmospheric circulation. The retreating sea ice cover and an associated warmer and fresher upper ocean have a direct effect on intensifying the stratification of the water column, with the potential to destabilize the thermohaline circulation, which regulates the Earth's Climate (Rahmstorf et al., 2002).'*

Introduction
- Address the significance of atmospheric conditions and climate patterns in relation to the retention of FWC in the Beaufort Gyre, which currently lacks explanation in the introduction.
*We have added this information in the following paragraph of the introduction:*

*'At the western side of the Arctic climate system lies the Beaufort Gyre, a large swirling circulation cell in the Beaufort Sea. The BG's rotation is driven by anticyclonic (clockwise) wind stress caused by a high-pressure system in the lower atmosphere. The gyre contains an enormous reservoir of sea ice and freshwater from northern rivers (mainly Mackenzie and Yukon) and the Bering Strait (Proshutinsky et al., 2015; Armitage et al., 2020). The shape and extension of the BG's is driven by weather patterns such as Arctic Oscillation (AO) and has a marked seasonal variability. Within the BG, freshwater accumulates through Ekman convergence, ultimately making its exit from the Arctic through the Davis and Fram Straits. Since 1997, high atmospheric pressure has triggered strong anticyclonic winds over the Beaufort Gyre (Lenton et al.,2019) which lead to an increase of FWC by 40% in the last two decades (McPhee et al., 2009; Solomon et al., 2021).*

*The variability of freshwater fluxes from the Arctic has the potential of collapsing subpolar North Atlantic convection, resulting in rapid North Atlantic cooling (Holliday et al., 2020) that would affect global climate via the thermohaline circulation (Rahmstorf, 2000; Zhang et al., 2021; Arthun et al., 2023), as well as the ocean heat content and biogeochemical cycles (Li et al., 2009). The timing and consequences of the eventual release of the accumulated freshwater into the Beaufort Gyre into the North Atlantic remain unclear and warrant further investigation.'*

- It might be best to describe what liquid freshwater content is since its not just a layer of freshwater on top of salt water, but a ratio of salt content that is lower than a certain standard. The salinity reference is also not agreed upon between scientists and may need to be defended on your part for why you chose that salinity reference (simply stating that you match the salinity reference with Proshutinsky et al. (2009) as you compare your results to their gridded FWC. It would also be beneficial to clarify the concept of FWC within the context of the Arctic. Emphasize that "freshwater" does not necessarily mean complete

absence of salinity; there exists a salinity threshold that varies in Arctic research.

*We have introduced the following phrase: 'The freshwater is defined as the amount of zero-salinity water that is contained in a volume of water relative to a reference salinity. Liquid freshwater content (FWC) is the depth integral of freshwater, expressed in length units. We chose the salinity reference used in Proshutinsky et al., 2009 as we will compare our estimations with their gridded in-situ estimates.'*

Line 17: Be more specific, the Arctic has experienced rapid changes more than just recent years, it has arguably been occurring over decades or at least since 2007. This statement could be improved by adding specific rate or timeframe that would enhance the importance of Arctic changes.

*We have changed the line by 'rapid changes in the last decades'*

Line 21: "(referred to as FWC)"
Change to "(FWC)"
*Done*

Line 23: Please provide a references for these contributions of FWC as it may differ between time periods or methodologies.
*Done, we have included the reference to Timmermans and Toole, 2023.*

Line 28: This sentence is a bit misleading, the Beaufort Gyre itself isn't necessarily at the 'heart' of the Arctic's climate, its highly influenced by climatic systems and atmospheric processes. It is also located in the Beaufort Sea and may extend northward but this statement makes it sound like it is not in the Beaufort Sea.

*We have rephrased as: 'At the western side of the Arctic'.*
*We have rephrased as 'in the Beaufort Sea'.*

Line 29-30: This is a personal opinion but will clarify for the reader: "…strong anticyclonic (counterclockwise) winds…powerful cyclonic (clockwise) circulation".
*We have rephrased the paragraph, now it reads as 'The BG's rotation is driven by anticyclonic (clockwise) wind stress caused by a high-pressure system in the lower atmosphere'*

Line 32-33: This transition between sentences seems random, you started introducing the Beaufort Gyre then talk about the Arctic's freshwater flux influence on a global scale without stating how. You could emphasize that the Beaufort Gyre consists of a large portion of the entire
Arctic Ocean's FWC and where the connection is between the Arctic and the thermohaline circulation comes in.
*You are right, sorry for the confusion. We have rewritten the entire paragraph.*

Line 35: I'm assuming "change" could be clarified as "increase"? This sentence also seems random and would benefit with explanation of the importance of the time between 2012 and 2016.
*Same as before, we have rewritten the paragraph.*

Line 40: Only within the last decade (2013-2023)?. Also, is sea surface height capitalized for a reason?
*Changed to: 'In the last decades, satellite data such as altimetry (e.g. sea surface height from CryoSat-2)..'*

Line 47-49: It would be much simpler to state that FWC is the integral of salinity differences between measurement and a constant. I understand the way you are introducing different ways of measuring FWC but you don't describe what FWC really is upfront. This paragraph can be conveyed more directly and concise.

*As stated in previous comment, we have introduced an explanation of FWC: 'The freshwater is defined as the amount of zero-salinity water that is contained in a volume of water relative to a reference salinity. Liquid freshwater content (FWC) is the depth integral of freshwater, expressed in length units. We chose the salinity reference used in Proshutinsky et al., 2009 as we will compare our estimations with their gridded in-situ estimates'*

Line 56-57: One-day repeat cycle does not seem accurate. Data can be generated at daily intervals, but that is not the same as a satellite's repeat cycle. Typically, its 3-8 days unless you can clarify what you mean. I also believe Aquarius's spatial resolution is much greater. The repeat cycle of SMOS in the Arctic is

*We have rephrased as: 'The SMOS satellite provides daily full coverage in polar regions with an effective spatial resolution of 50 km in the seasonally ice-free areas of the Arctic Ocean.'*

Line 62: Change "sea surface salinity" to "SSS". Also in lines 70, 228.
*Done*

Line 86: Change "resolution" to "spatial resolution"
*Done*

Line 90: "ranging from surface to bottom". Can you be more specific on what 'surface' is in meters?
*We have rephrased: 'ranging from surface (zero meters) to bottom.'*

Data
Line 70: State the level of the satellite product (level 3 product?)
*Done*

Line 71: Specify month and year range of SMOS data and which timeframe you took the data from here.
*Done*

Line 80-81: You use the OSI-SAF acronym then the name but should it come after "Ocean and Sea Ice Satellite Application Facility" in parenthesis?
*Changed*

Line 96: Change "sea ice concentration" to SIC. Also in line 139.
*Done*

Line 99: "..surface to 4000 m depth.", is this near surface or skin-surface? Might be best to note top layer in meters.
*Now it reads as: 'It has 40 hybrid vertical layers (z-isopycnal) from the surface (0 m) to 4000 m depth with resolution varying from 1 m at the surface to 1500 m at the deepest level.'*

Line 105: Do the in-situ data exceed 300 meters as well or is this referring to the region's depths exceed 300 m defined by Proshutinsky. If the latter, then clearly define the depths that the in-situ measurements cover.

*We refer at how the Beaufort Gyre area is defined. The details of in-situ measurements used to produce this FWC product is in the paper of Proshutinsky et al., 2009, and we have cited it in this subsection. For example, the ITPs used go up to 760 meters and the moorings up to 2000 meters.*

Line 109: Specify the time intervals that these data are provided or that you used (hourly, daily, monthly?)

*To clarify this part, we have included the following detail: 'They offer a yearly estimate based on those in-situ measurements from July to October.'*

Eq.1: Define equation symbols through text explanation.

*We have added: 'Where S is the salinity at each grid point, $S_{ref}$ is the salinity reference, and z is the depth where the $S_{ref}$ is achieved.'*

Line 121: Justify the use of 16, 25, and 29 meters for the MLD. Can you use profile data to estimate the MLD? You mention using a constant value of MLD throughout the year but are only assessing melt season, which is confusing.

*As responded in the general comment, we have added an explanation and further details in subsection 2.4.*

Results

Line 131: As a predominantly anticyclonic gyre, isn't it more characteristic for the Beaufort Gyre to have downwelling occur instead of upwelling as this line suggests (i.e. "surfaced")? Or are you discussing waters pulled from the surrounding water or characteristics in general?

*This is a general characteristic of SMOS capability but as is misleading in this context, we decided to delete this sentence.*

Figure 1: Why these two years (2011 & 2016)? You mention the greatest and quickest change in salinity between 2012-2016, but do you choose 2011 since is the beginning of the period you are examining?

*We have chosen these two years because they were an example of clear qualitative melting signatures in SMOS SSS. We did delete the phrase concerning maximum salinity anomalies in years 2011 and 2016 as they referred to the entire Arctic region and it was confusing.*

Stating the uppermost level of TOPAS4b would be helpful in analyzing the difference between the skin-salinity that the satellite measures and the depth of what TOPAZ4b measures.

*We have already introduced the depth of first layer defined in TOPAZ4b system.*

Line 137: Should the references be in parentheses?
*Done*

Line 155-156: Repeat of the description of the Beaufort Gyre region as in lines 104-105, could further describe why you use the same region as Proshutinsky et al.

*Ok, we stated that we used the same area.*

Figure 3. Note the acronym for freshwater content and its unit in the caption (i.e. FWC [m]). The colorbar for (c,f) seems to emphasize regions over or under a ~2psu difference, is there a

reason for this or can the colorbar be changed to detail the region and make different values clearer?

*We have changed all the captions. We decide to truncate the anomaly values higher than 3 psu to empathize the biggest contribution in areas affected by sea ice melting.*

Line 182: Parenthesis around reference.
*Done*

Line 182-184: How does TOPAZ4b assimilate variables associated with freshwater inputs? Are there estimations on its certainty to capture these signals accurately?

*TOPAZ4b used monthly climatological river inputs (as detailed in Xie et al. 2017) which were overestimated in Winter and underestimated in the Spring. The only relevant data assimilated were sea ice thickness from November 2010 onwards, but there were otherwise only sporadic salinity profiles in the area of interest.*

Section 3.2
Figure 5. Label [b] colorbar as "FWC error [m]". Last sentence of caption is not complete.
*We have changed label of 5b. Last sentence is finished.*

Line 198: You mention "several studies" but only cite one, can you provide more studies to back this claim? Otherwise reword.
*Done*

Figure 6.: Define more reasons why there are large discrepancies in FWC between in-situ data and other products. Could in-situ be overestimating due to lack of spatiotemporal coverage? The in-situ based FWC product is integrated so this could pose it's own errors or overestimation/underestimation.

*Thank you, we have included the following phrase: 'The reasons why in-situ estimates may overestimate FWC could be explained by the lack of spatiotemporal coverage of these measurements or by the fact that it is an integrated product with associated errors.'*

Line 228: Change "sea surface salinity (SSS)" to "SSS"
*Done*

Line 230: Change "…freshwater content.." to "FWC"
*Done*

Lines 251-252: "This integrated approach could allow for the identification of patterns, trends, and anomalies in SSS.."
*I'm not entirely sure I understand what you mean by that.*

---

## Author Comment (AC2)

**Reply on RC2**

*Dear reviewer, your insightful review is greatly appreciated. All of your suggestions have been tacked into account.*

**General comments:**

This study aims to estimate the improvement of liquid freshwater content by adding satellite sea surface salinity to reanalysis salinity at depth in the ice-free region of the Beaufort Sea. The analysis combines the salinity data from the TOPAZ4b reanalysis at various depths with the SMOS SSS values for the layers above the three specified fixed mixed layer depths. The authors suggest a clear improvement in the liquid freshwater content estimation when adding satellite sea surface salinity above the fixed mixed layer depths, especially in areas close to ice melting.

The authors' idea of adding the salinity satellite data to the reanalysis in estimating the high latitude liquid freshwater content is rational and vital for monitoring one of the key processes - the large-scale changes in Arctic freshwater content. However, there are some validation and methodological issues to the calculation of liquid freshwater content that require further examination. The manuscript should be suitable for publication once these issues are clarified.

**Specific comments:**

Line 10-11. There are many different definitions for calculating mixed layer depths. A sharp halocline near the sea surface could "create" a shallow surface mixed layer that is much shallower (but more realistic) than a "conventional" mixed layer depth that is estimated by bulk SST and SSS measurements and deeper profiles. Therefore, it would be very helpful to say something like "when adding satellite sea surface salinity above the mixed layer that is calculated by deeper/bulk SSS".

*We have rephrased this phrase as: 'The results highlight the underestimation of the freshwater content using reanalysis data in the Beaufort Sea and a clear improvement in the freshwater content estimation when adding satellite sea surface salinity measurements **in** the mixed layer.' The information on how the MLD is calculated is detailed in paper by Toole et al., 2010 cited in section 2.4.*

Line 29: Lenton et al. 2019 may not be suitable for the statement: "since 1997, high atmospheric pressure has triggered strong anticyclonic winds over the Beaufort Gyre area".

*We have deleted the reference in this line.*

For section 2.4. Are you considering or evaluating the feasibility of combining SMOS SSS and in-situ ocean salinity data (e.g., CTD, XCTD, and UCTD) mentioned in section 2.3?

*In section 2.4 we explain how we combine SMOS SSS with TOPAZ4b to compute FWC, in section 2.3 we explain the in-situ based estimation of FWC produced as described in Proshutinsky et al 2009. For the moment we are not considering to combine in-situ with SMOS SSS but we will consider your suggestion for future research.*

Figure 1: please clarify the mean uppermost salinity level of TOPAZ4b used at least in the figure captions. The uppermost salinity level from

https://data.marine.copernicus.eu/product/ARCTIC_MULTIYEAR_PHY_002_003/description is 0m. Is it the level of TOPAZ4b in this study?

*Yes, this is the level used, we have clarified this point in section 2.2 when presenting TOPAZ4b*

Line 157-158: Please clarify if the authors combine the salinity data from the TOPAZ4b reanalysis at depths above the MLD and only replace the surface level of TOPAZ4b salinity with SMOS SSS values for computing the freshwater content. It would be beneficial to state it in section 2.

*We combine the salinity data from the TOPAZ4b reanalysis at depths bellow the MLD and replace the levels of TOPAZ4b above MLD with SMOS SSS values in areas free of ice for computing the freshwater content, we explain this point in Section 2.4.*

Also in the caption of Figure 3 (also the paragraph starting in Line 160) the author states that "The freshwater content difference is computed as freshwater content from TOPAZ4b salinity minus the freshwater content from TOPAZ4b adding SMOS up to 16 meters.". Could the authors please clarify if the freshwater content in Figure 3 (b) and Figure 3 (e) is calculated. What does adding SMOS up to 16 meters mean? Does it replace all the TOPAZ4b salinity above 16 m with SMOS SSS or just replacing the top level (0m?) of TOPAZ4b salinity to SMOS SSS?

*It is replacing salinity values of TOPAZ4b with SMOS SSS values in all levels of TOPAZ4b up to 16 meters.*

The main concern is that SMOS SSS may be representative for the very fresh but thin of the surface layer from sea ice melting. SMOS SSS could possibly be representative for less than the upper 5 m of the surface layer. Could the authors plot vertical profiles merging "SMOS SSS as the surface salinity value" and "the collocated in-situ profiles (the uppermost level shallower than 2~3 m would be great or at least shallower than 10 m)" in the area to confirm if this method is representative to the vertical structure. Toole et al. (2010) that is cited in section 2.4 and 3.1 mention (in their section 2.1) that their ITPs were programmed to sample the water column between 7 and 750m. Many of their ITP profiles in midsummer indicate that the Canada Basin ML is frequently thinner than 10m. Their abstract also suggests that "The July–August mean mixed layer depth based on the Ice-Tethered Profiler data averaged 16 m (an overestimate due to the Ice-Tethered Profiler sampling characteristics and present analysis procedures)". The point is, even though replacing the entire 16, 25 or 29 m of TOPAZ4b salinity to SMOS SSS increases the FWC estimation, this method likely overestimates the contribution of SMOS SSS. The deviation of overall near-surface thermohaline structure of TOPAZ4b/ reanalysis from the observations may contribute to much of the underestimation of the FWC compared to in-situ measurements.

*Indeed, the reason why the reanalysis system underestimates FWC compared to in-situ measurements not only lies in the near-surface thermohaline structure but may also be affected by the use of a river climatology that underestimates discharge or coupled with an ice model that underestimates ice thickness. As you point out, it is possible that our method may overstate the importance of surface salinity by adding SMOS salinity throughout the entire mixed layer. However, it is clear from our experiments that integrating the surface salinity observed by SMOS in the upper ocean substantially improves the freshwater depth in the Beaufort Gyre.*

*In this work we aim to emphasize the value of satellite surface salinity data in the study of FWC. Sure, we make assumptions about how representative surface salinity is of the mixed layer. We acknowledge that the depth of the mixed layer can be further refined, and we want to address this point in the near future. We account for this source of uncertainty conducting the tests with three different mixed layers to account for the spatiotemporal variability of the MLD, and the errors associated to our need to integrate surface salinity using a fixed-layer model.*

Line 206: Please specify what the slope is indicated. Is this the slope estimated in Figure 7? It is better to specify what the slope is in Line 225 as well.

*Yes, it is related to Figure 7, we have integrated this information in the text.*

Line 215 states that "the results show a significant improvement in terms of bias" but in figure 7 the biases on the two figures are both 1.81. Please clarify.

*The bias on the Figure 7a is 1.81 and the Figure 7b is 1.18*

Line 219: It is not very clear what dispersion is. Is it the difference between the estimation of TOPAZ only or TOPAZ+SMOS SSS? Please specify. Also, it is not very clear what "the dispersion remains stable" means. Does it mean the difference between the two (not sure what the two are) does not change with time or ?

*The dispersion refers to the standard deviation, and the error remains stable, refers in the three experiments, we have clarified that in the text.*

**Technical corrections:**

Line 53: SMAP should be launched and has become operational since 2015.

*Changed*

Line 137: Might want to add parentheses to the citations.

*Done*

Line 137-139: This sentence "Note that even if TOPAZ4b reanalysis assimilates SMOS SSS, the resulting surface salinity does not seem to reproduce the same SSS dynamics as seen by SMOS." seem to belong to the next paragraph. Please consider reconstructing these two paragraphs.

*Changed*

Figure 3 captions: Please move (a,d) (b,e) (b,e) (c,f) (top row) (bottom row) to before the items described.

*Done*

Table 1: Please consider an option of adding the values in table 1 to an additional subfigure of Figure 4 for easier reading instead of stating the yearly mean in numbers. Even though it will

be only about 10 data points for each fwc estimate, it would be easier to visualize the yearly mean variation and the differences between different fwc estimates.

*If I understand correctly, the information you're requesting is already included in Figure 6, where we have also incorporated the estimation from in-situ data.*

---

## Referee Report (RR1)

Reviewer's Comments:

**Technical Revisions:**
Lines 6-9. Sentences can be combined or reworded shorter

Line 13 "…this key process that is creating subtle density differences that have the potential". What key process? The revision does not clearly answer the previous comment made. Possible change to "…understanding key processes related to salinity variations that cause density differences with potential to influence the…"

Line 24: Add period to end of sentence.

Line 25: 'Freshwater' in the ocean is NOT zero-salinity, this is confusing for the reader. Reword to emphasize that it is salinity less than the reference salinity.

Line 27: State the reference salinity here to be direct.

Line 61: "Sea Surface Salinity" remove capitalization.

Lines 126-127: Remove period after "psu" in equation (1) and start the sentence "Where" with uncapitalized 'w'.

---

## Author Response (AR2)

Anonymous referee #1. Report #2

Technical Revisions:
Lines 6-9. Sentences can be combined or reworded shorter

Done, now the sentence reads as: 'In this work, we evaluate the freshwater content in the Beaufort Gyre using surface salinity measurements from the satellite radiometric mission Soil Moisture and Ocean Salinity (SMOS) and TOPAZ4b reanalysis salinity at depth, estimating the freshwater content from 2011 to 2019 and validating the results with in-situ measurements.'

Line 13 "…this key process that is creating subtle density differences that have the potential". What key process? The revision does not clearly answer the previous comment made. Possible change to "…understanding key processes related to salinity variations that cause density differences with potential to influence the..."

Applied, thank you.

Line 24: Add period to end of sentence.

Done.

Line 25: 'Freshwater' in the ocean is NOT zero-salinity, this is confusing for the reader. Reword to emphasize that it is salinity less than the reference salinity.

We don't understand what you mean. For us the definition is correct, the quantity of freshwater that is needed to add to an area to reach the reference salinity.

Line 27: State the reference salinity here to be direct.

Done.

Line 61: "Sea Surface Salinity" remove capitalization.

Done.

Lines 126-127: Remove period after "psu" in equation (1) and start the sentence "Where" with uncapitalized 'w'.

Done.
* * *
Anonymous referee #2. Report #1

General Comments:

The main concern still lies on the fixed mixed layer depth being used in the analyses. The fixed mixed layer depths cited (Toole et al. 2010) is estimated by bulk profiles from 7 m and below. The manuscript aims to improve the FWC estimation using the satellite measured surface (~1 cm or less) values as the salinity above the fixed mixed layer depths. The upper ocean structure near the ice edge, river runoff… could be quite different from swapping all salinity above a fixed depth to SMOS SSS. Even though the results suggest that integrating the SMOS SSS in the upper ocean improves the freshwater depth in the Beaufort Gyre, the uncertainties from using the fixed mixed layer depths for the entire region could be problematic. Below are some possible suggestions that may help support your estimation not being blurred by large uncertainties in overly simplified methodology.

(1) Estimate mixed layer depths. The mixed layer depths can be estimated using the reanalysis data. Compare the distribution of the varying mixed layer depths with the fixed mixed layer depths. If the distribution is highly centered and close to one of your fixed mixed layer depths, it supports your results. And this may inform how much uncertainty comes from using fixed mixed layer depths alone. If the distribution is broad, then it would be beneficial to use "calculated" mixed layer depths instead. For example, the mixed layer depths could be deeper in parts of the Beaufort Gyre (as you mentioned in Line 235-236) while some closer to ice edges/ river runoff/ freshly melt waters could be very shallow (though this may or may not be well represented in the reanalysis data).
Even though the FWC results from estimating the mixed layer depths may or may not change your conclusions substantially, the value of the results would be much improved by using a more reliable method.

To ensure the coherence of the MLD estimations utilized (Toole et al., 2010), we conducted an analysis of in-situ vertical salinity profiles in the Beaufort area for the summer of 2019.

As depicted in the figures below:

[Figure]

Several in-situ salinity casts were obtained in September, indicating that assuming an MLD between 15 and 30 meters aligns coherently with structure revealed by the in-situ data.

Specific comments:

Line 105-106: from in-situ source? Please specify.
Done.

OSISAF sea ice concentrations are plotted in Figure 1 and Figure 2 but not mentioned in the data section. Please include it in the data section.

OSISAF data is described in lines 89-91.